# Features of DNA–Montmorillonite Binding Visualized by Atomic Force Microscopy

**DOI:** 10.3390/ijms24129827

**Published:** 2023-06-06

**Authors:** Sergey V. Kraevsky, Nikolay A. Barinov, Olga V. Morozova, Vladimir V. Palyulin, Alena V. Kremleva, Dmitry V. Klinov

**Affiliations:** 1Federal Research and Clinical Center of Physical-Chemical Medicine of Federal Medical Biological Agency, 1a Malaya Pirogovskaya Street, 119435 Moscow, Russia; 2Alikhanov Institute for Theoretical and Experimental Physics, National Research Center “Kurchatov Institute”, ac. Kurchatov, sq, 1, 123182 Moscow, Russia; 3Institute of Biomedical Chemistry, Pogodinskaya Street, 10, Build 8, 119121 Moscow, Russia; 4Moscow Institute of Physics and Technology, 9 Institutsky Per., 141700 Dolgoprudny, Russia; 5National Research Center of Epidemiology and Microbiology Named after N.F. Gamaleya, Ivanovsky Institute of Virology of the Russian Ministry of Health, 16 Gamaleya Street, 123098 Moscow, Russia; 6Applied AI Center, Skolkovo Institute of Science and Technology, Bol’shoy Bul’var, 30, bld 1, 121205 Moscow, Russia; 7Department Chemie, Technische Universität München, 85748 Garching, Germany

**Keywords:** montmorillonite, DNA, atomic force microscopy, edge joints, density functional theory

## Abstract

In the present work, complexes of DNA with nano-clay montmorillonite (Mt) were investigated by means of atomic force microscopy (AFM) under various conditions. In contrast to the integral methods of analysis of the sorption of DNA on clay, AFM allowed us to study this process at the molecular level in detail. DNA molecules in the deionized water were shown to form a 2D fiber network weakly bound to both Mt and mica. The binding sites are mostly along Mt edges. The addition of Mg^2+^ cations led to the separation of DNA fibers into separate molecules, which bound mainly to the edge joints of the Mt particles according to our reactivity estimations. After the incubation of DNA with Mg^2+^, the DNA fibers were capable of wrapping around the Mt particles and were weakly bound to the Mt edge surfaces. The reversible sorption of nucleic acids onto the Mt surface allows it to be used for both RNA and DNA isolation for further reverse transcription and polymerase chain reaction (PCR). Our results show that the strongest binding sites for DNA are the edge joints of Mt particles.

## 1. Introduction

Various biotechnological, medical and nanotechnological applications benefit from easy-to-process, relatively cheap and easy-to-handle raw materials. Clay minerals are a perfect example of such materials being abundant in nature and, hence, are widely used as sources of nanoparticles [1,2] and as additives in nanocomposites [3,4]. They also show a biocompatibility at substantially higher doses compared to other nanomaterials [5]. DNA represents another important “building block” used in nanotechnological research and industry due to its versatility [6], which creates the possibility to obtain particular three-dimensional shapes reliably via its synthesis [7,8], and its importance as one of the main molecules of life. The first studies of DNA–clay interactions were motivated by Bernal’s hypothesis of the prebiotic evolution of nucleotides on the non-organic replication surfaces of silicates [9]. It was shown that the polymerization of RNA could be catalyzed by montmorillonite (Mt), a common member of the smectite group of clay minerals [10]. The hypothesis was also supported recently by a synthesis of clay vesicles which, as the authors of [11] speculate, could have been the first cell-like compartments. Moreover, the interactions of Mt particles and nucleotides are capable of protecting nucleotides from formamide-mediated destruction [12] as well as preventing nucleic acids from being cut by nucleases [13,14,15]. Mt particles were shown to preserve the DNA in the acidic environment of mouse stomachs [16], which strengthened the research that considers clay-based gene delivery systems [17,18,19]. In particular, Mt appeared to be one of the simplest gene delivery systems transferring plasmid DNA into eukaryotic cell lines [17]. In the latter case, for the effective cloned gene expression in transfected eukaryotic cells, the desorption of DNA from Mt was necessary. As was shown in Ref. [20], Mt can be used for DNA purification, which is essential for transfection, sequencing and PCR in biomedical applications. RNA when immobilized on the Mt surface is also more stable than free RNA in a water solution and has been shown to serve as a template for reverse transcription with a PCR experiment [20]. The same experiment revealed that Mt did not inhibit the enzymes.

Mt surfaces were shown to favor the adsorption and reactivity of mono- [21] and polynucleotides [17,20]. However, despite the comprehensive analysis of binding between the nucleotides and Mt under ambient conditions in a broad range of pH and in the presence of various salts [21,22,23], the adsorption of RNA and DNA cannot be guaranteed due to their instability at extreme pH levels far from physiological conditions.

An investigation of clay–DNA complexes cannot be comprehensive without using different types of direct visualization techniques. Transmission electron microscopy (TEM) and scanning electron microscopy (SEM) images show the presence of long DNA strands that, along their length, appear to be attached to several points at the edges of exfoliated clay particles [24]. Between these points of contact, they often spread, forming large loops [25].

Atomic force microscopy (AFM) has some advantages over electron microscopy methods. In particular, AFM is capable of measuring adhesive and mechanical properties at the molecular level and studying samples in different surroundings without pretreatments such as microtoming or metal coating, which sometimes lead to changes in sample properties. Therefore, the AFM approach presents new opportunities for studying clay–DNA complexes.

The most frequently used substrate for systems with DNA by means of AFM measurements is mica. It has a flat structure, and DNA binds to it in the presence of millimolar concentrations of divalent metal counterions (e.g., Mg^2+^) [26,27,28,29,30,31,32]. Direct measurements of the binding energy between phosphate/pyridine-modified cantilever and mica surfaces show that the edge surfaces of phyllosilicates are more reactive to phosphate compared to the basal ones [33].

When DNA comes in contact with clay particles, bonds of different strengths can form [34]. The DNA adsorption on the basal surface of clay occurs via weak bonds, whereas the strong bonds form when DNA interacts with the clay edges [35,36]. The chemical properties of amphoteric groups at the edge surfaces and the free energy of the specific adsorption of cations (Ni^2+^, UO_2_^2+^, etc.) are dependent on the crystallographic orientation of the surface [37] and are well studied by ab initio molecular dynamics (MD) as well as quantum chemical computer simulations [38,39,40,41,42,43,44,45,46,47,48,49,50].

In the following parts of the present study, we show and analyze AFM images of DNA complexes with Mt at the mica support at various conditions. The images allowed us to identify the main binding sites in agreement with our estimations based on density functional theory (DFT) calculations.

## 2. Results and Discussion

### 2.1. DNA and Mt in Distilled Water

The AFM technique is very sensitive to sample impurities. Thus, the visualization and further interpretation of AFM images can be greatly impeded by the addition of salts or organic ligands to the system. We first present the images for the simplest treatment (no salt added) and then describe the cases with more complex preparation procedures.

Figure 1a shows the AFM image of the reference sample (see Materials and Methods). The DNA aggregates on mica [27] and forms fibers, which are stretched between Mt particles and weakly bound to the basal surface of mica support (Figure 1a). These DNA fibers can reach 40 nm in width and 2 nm in height (Figure 1c). The height of the fibers exceeds the thickness of Mt (~1 nm). DNA wraps around Mt particles bound to the edges and, thus, links clay particles into a network with the fibers working as connections on the surface of the mica support. The DNA fibers seem to avoid the clay basal surfaces, possibly due to electrostatic repulsion. A similar result was shown with electron microscopy imaging in the study of Khanna et al. [25], in which DNA complexes with kaolinite and Mt were investigated (see Figure 2 in Khanna et al. [25]). Interestingly, the authors observed both fibers and separate double-stranded DNA molecules bound to kaolinite hexagonal-shaped particles. Mt, in turn, exhibited a flake-like appearance, which made it rather difficult to visualize the bound DNA strands and the corresponding binding sites. In the present study, we could prepare and visualize single-layer well-shaped particles of Mt, which allowed us to detect the binding sites of DNA (Figure 1 and Figure 2). DNA-Mt complexes at pH 3.5 and pH 9 are shown in Appendix A (Appendix A, respectively).

### 2.2. DNA–Mt Interaction in a Mg^2+^ Solution

#### 2.2.1. Rinsed DNA and Mt Samples with Mg^2+^ Solution

The effect of salinity on nucleotide sorption on clays was inspected earlier in the literature [51]. The increase in the concentration of Mg^2+^ cations was shown to improve the sorption of deoxyguanosine-5′-monophosphate (dGMP) on nontronite and pyrophyllite. We investigated the effect of Mg^2+^ cations on the binding of DNA to Mt. The reference sample was washed with a MgCl_2_ solution for 1 min. The details of the procedure can be seen in Materials and Methods. The AFM images of the initial (without Mg^2+^ cations) and resulting (with Mg^2+^ cations) systems are shown in Figure 2a,b, respectively. Adding Mg^2+^ cations to the system clearly changes the binding between Mt/mica and DNA. In the presence of Mg^2+^, the aggregated DNA fibers unfold into separate molecules; compare Figure 2a,b. In Figure 2b (after washing with MgCl_2_), the DNA strands lie separately from each other, and no network is formed. The height of the separate DNA strands is about 1 nm, which is only half of the height measured for the fibers (see above) because the measurement shows the value relative to the salt layer rather than the surface. The width is also smaller than for the fibers. To note, the thickness of individual double-stranded DNA molecules is ca. 2 nm [52]. Interestingly, the measured width of the molecules in this case is about 15 nm, most probably due to the influence of a salt layer surrounding the DNA molecule [53]. In Figure 2b, one also can notice that most of the DNA strands are attached to the edge joints of the Mt particles. However, there are a few DNA strands attached to the edge surfaces, as shown in the upper left corner in Figure 2b. Figure 2c–e show magnified pictures of the selected square areas from Figure 2b. One can see that the DNA strands attach with their one end to the edge or edge joint of the Mt particle. Figure 2c clearly shows the same DNA strand binding twice to the edge of the Mt particle. Our observation agrees well with the assumption of Yamaguchi and coauthors that DNA immobilized on Mt could form protruding DNA loops [54]. The loops were also observed in TEM images by Franchi [25], in which DNA threads were forming bonds with several points on the edges or on edge joints of Mt, from which they spread in a loop. The presence of bundles in between the Mt particles and mica (Appendix A) can be considered as an indicator of possible particle stacking and liquid phase formation as in [54].

#### 2.2.2. DNA and Mt Mixed in Mg^2+^ Solution

In the experiments shown in Figure 2, we added Mg^2+^ after the DNA was spread over the surface. Alternatively, Mg^2+^ can be added to the system before DNA is present (see Materials and Methods). The AFM images of the resulting sample are shown in Figure 3. Extra AFM images are provided in the Appendix A. Again, it can be seen that in the presence of magnesium ions, the DNA fiber on the mica is unfolded into separate double-stranded DNA molecules. However, the DNA filaments are still seen on the basal surface of Mt particles as shown by the bright region on the surface of the Mt particle in Figure 3a. For the separate DNA single strands, the height is measured at 1 nm, as in the previous case described above. The width of the strands is measured up to 8 nm, which is comparable to that seen in Figure 2b (see above).

We show the magnification of some areas where DNA is bound to Mt in Figure 2 and Figure 3 in order to see the sites at which DNA strands are bound. The zoomed images in Figure 2c–e and Figure 3b,c clearly show that the DNA binds at the corner junctions of the edges. However, there are a few bonds at the edge surfaces as well; see Figure 2c and Figure 3c.

In order to confirm that the DNA mainly interacts with Mt edges, a series of control experiments with a topologically similar 2D material, graphene oxide, was performed. The detailed description of these control experiments is given in the Appendix A. Appendix A clearly shows a few DNA bundles lying on the basal surface of the graphene oxide particles. Thus, it was concluded that the observed DNA bundles were capable of wrapping around the Mt particles and were weakly bound to the Mt edge surfaces.

### 2.3. DFT-Based Interpretation of Edge Joint Reactivity

The most frequently discussed mechanism of nucleotides adsorption on phyllosilicates is binding via a phosphate group ([21,22,51] and references therein). Several studies of the adsorption of dissolved phosphates, nucleic acids or organic matter on minerals have shown that phosphate could form mono- or bidentate inner-sphere complexes with the metal of the surface via a ligand exchange mechanism ([21,22,51] and references therein). The surface OH groups are exchanged by phosphate groups with the release of water molecules. In the case of Mt particles, such complexation is possible via dangling bonds, which mostly belong to edge surfaces or are located at the edge joints. Since ionic Al-O bonds are usually weaker than covalent Si-O bonds [55], the phosphate groups are expected to bind to Al centers rather than to Si. It is intuitively clear that the density of dangling surface groups defines the reactivity of the corresponding surface’s metal center because dangling oxygen centers can undergo protonation, and, as a result, OH or OH2 ligands can be exchanged with other ligands or water molecules in a solution. The most common edge surfaces of 2:1 clay minerals predicted by crystal growth theory and observed in experiments are (010) and (110) [56,57,58,59,60,61]. Each of the edges has different chemical groups at the surface as shown in Figure 4. Both surfaces were extensively studied earlier [38,39,62]. The dangling bonds of surface Al for the neutral ideal (010) edge surface are Al-OH2 and Al-OH. For the ideal (110) edge surface, each Al exhibits only one dangling bond, Al-OH2, see Figure 4.

Due to the pseudohexagonal structure and local symmetry of 2:1 phyllosilicates, the edge joints occur either between the (010) and (110) edge surfaces or between the (110) and (110) edge surfaces with the crossing angle being either 60° or 120° (Appendix A for details). Using the structures of the (010) and (110) surfaces known from the literature [38,39], one can construct the corresponding edge joints, as shown in Figure 5. The dangling oxygen (O) surface groups are shown by arrows. The blue ones correspond to the edge surface groups while the red ones correspond to the groups on the edge joints. We see that three is the maximum number of red arrows pointing to a single Al center. This center is located at the edge joint between the (010) and (110) surfaces. In order to roughly estimate the reactivity of various Al centers, we performed a DFT-based calculation of OH2 ligand dissociation energy from four Al centers, which are shown by numbers 1 to 4 in Figure 5. Numbers 2 and 4 correspond to the Al centers on the (110) and (010) edge surfaces, while 1 and 3 correspond to the (110) × (110) and (110) × (010) joints, respectively. The DFT calculations were performed using the plane-wave-based Vienna ab initio simulation package (VASP) [63,64,65,66], and the details are described in the Appendix A. Estimated OH2 dissociation energies are lowest for the joints, with 4 and 2 kcal/mol for sites 1 and 3, respectively. The energies of 16 and 11 kcal/mol were estimated for the OH2 ligand dissociation from Al centers 2 and 4, respectively (Appendix A). Thus, one expects joints to be more reactive compared to the edge surfaces. Additionally, the edge joints are expected to exhibit smaller steric hindrance for the adsorbed DNA fragment compared to more or less flat edge surfaces.

The isolation of nucleic acids by Mt may be used for biomedical applications such as RT-PCR. A direct comparison of highly purified Mt and commercially used silica gel showed comparable results (Appendix A). Currently, silica nano- and microstructures are widely applied for industrial and household applications (such as food, beer production, and cosmetics) as well as for clinical purposes in diagnostics and pharmaceutical products [67]. Because of their small sizes and large surface area, silica nanostructures exhibit unique bioactivities and are capable of interactions with cellular or subcellular structures. The binding of SNS and Mt with nucleic acids may be used for molecular diagnostics by means of reverse transcription with real-time PCR, ligase chain reaction, loop-mediated isothermal amplification (LAMP) and the molecular hybridization of nucleic acids. Importantly, DNA and RNA can be adsorbed and desorbed from the Mt surface at different osmotic conditions (see the section titled PCR experiment in the Appendix A and [20]). The biocompatible and safe nanomaterials with DNA-binding capacities are also necessary for gene delivery applications such as gene immunization and therapy [68]. Non-viral vectors have been preferred due to their ease of production, controlled chemical composition, high chemical versatility, biosafety issues, and low immunogenicity and constitute an ideal alternative to viral vectors.

## 3. Materials and Methods

### 3.1. Sample Preparation

The Kunipia-P sample used in this study is a highly purified Na- Mt produced by Kunimine industries Co., Ltd. (Tokyo, Japan). It contains nearly 100% Mt [61,69,70]. Kunipia-P Mt was dispersed in deionized water at concentration of 0.01 g/L and ultrasonicated for 10 min at 30 °C and 60 W before the sample preparation. We used phage Lambda DNA (Fermentas Ltd., Burlington, ON, USA) that was 48,502 bp long.

The reference sample was prepared in the following way: 1 μL drop of the dispersion with Mt was placed on the surface of freshly cleaved mica, and 1 μL drop of the DNA solution was added to it for 1 min. Then, the reference sample was softly rinsed by placing a 100 μL drop of double distilled water for 1 min, which was then removed from the surface with a nitrogen flow. To avoid the formation of aggregates, Mt and DNA were not mixed directly in the bulk but rather in the vicinity of the surface. Thus, we could see the complexes of Mt and DNA with Mt particles not overlapping with each other.

The effect of Mg^2+^ cations on DNA binding to Mt was studied in two ways. First, the reference sample was softly rinsed with a 100 μL drop of 10 mM MgCl_2_ for 1 min. The water was removed from the surface by a nitrogen flow. Second, after placing 1 μL drop of the Mt solution on the surface of mica, 1 μL drop of the 10 mM MgCl_2_ was added, and, then, 1 μL drop of the DNA was added for 1 min. Then, the sample was softly rinsed by placing a 100 μL drop of double distilled water for 1 min, which was removed from the surface with a nitrogen flow afterwards.

### 3.2. AFM

AFM images were obtained in air using multimode atomic force microscope Ntegra Prima (NT-MDT, Russia) operated in intermittent contact mode. Ultrasharp AFM probes (carbon nanowhiskers with a curvature radius of several nanometers grown at tips of common silicon cantilevers) with a spring constant of 5–30 N/m were used [71]. The scanning was performed with small cantilever oscillation amplitudes within 3–10 nm. The repulsion regime was used for the acquisition of the highest possible AFM resolution in the operation with the ultrasharp cantilevers; the operational protocol is described in more detail by Prokhorov et al. [72]. The AFM images were analyzed using the Femtoscan Online software (www.nanoscopy.net/en/Femtoscan-V.shtm (accessed on 6 May 2021)).

## 4. Conclusions

Our AFM visualization shows that DNA demonstrates diverse behavior while interacting with Mt nanoparticles. If both Mt nanoparticles and DNA are deposited on the mica via a water solution, the DNA forms a network of filaments (bundles), which loosely binds to the Mt particles. If the system of already adsorbed DNAs on Mt on mica support is treated with Mg^2+^ cations it leads to the splitting of DNA bundles into separate DNA molecules and the network of filaments disappears. We have identified and computationally rationalized the preferential attachment of DNA to the edge joints of Mt particles in a wide pH range. However, a few chains are still attached to the edge surfaces. We observe that, between the points of attachment, the DNA forms loops, which is in agreement with earlier TEM studies [25].

The sample preparation appears to be important. If one adds the magnesium ions prior to the introduction of DNA to the system, the adsorbed DNA unfolds into separate molecules. However, if the magnesium cations are added later, the DNA ribbons on the basal surface of the Mt nanoparticles are preserved. Indeed, our DFT-based calculations show that the edge joints seem to be more reactive and, thus, more likely to be a preferential point of DNA attachment to Mt.

## Figures and Tables

**Figure 1 ijms-24-09827-f001:**
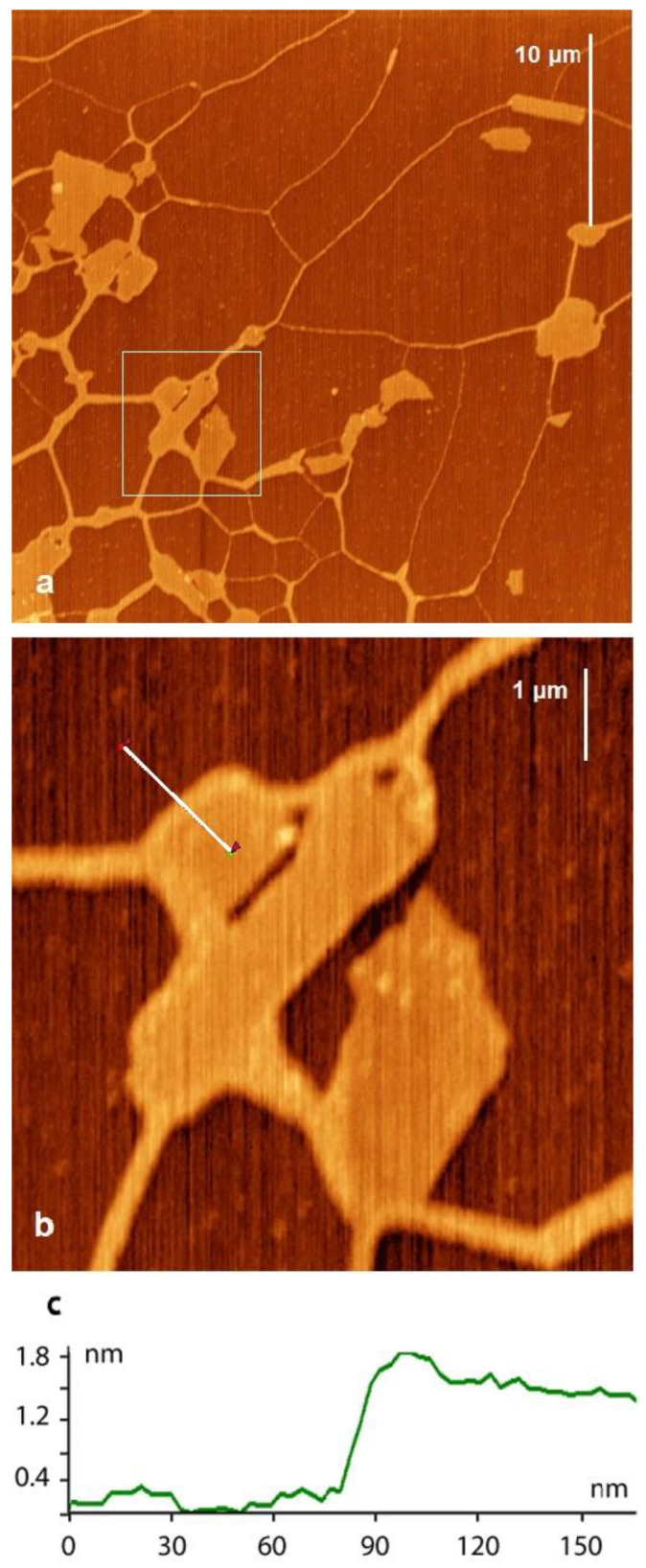
(**a**) AFM image of the mixture of DNA and Mt deposited on the surface of the mica. (**b**) Detailed view of the selected area. See the white square in (**a**). (**c**) AFM profile along the white line on the AFM image (**b**). DNA and Mt were mixed in deionized water at pH 5.8.

**Figure 2 ijms-24-09827-f002:**
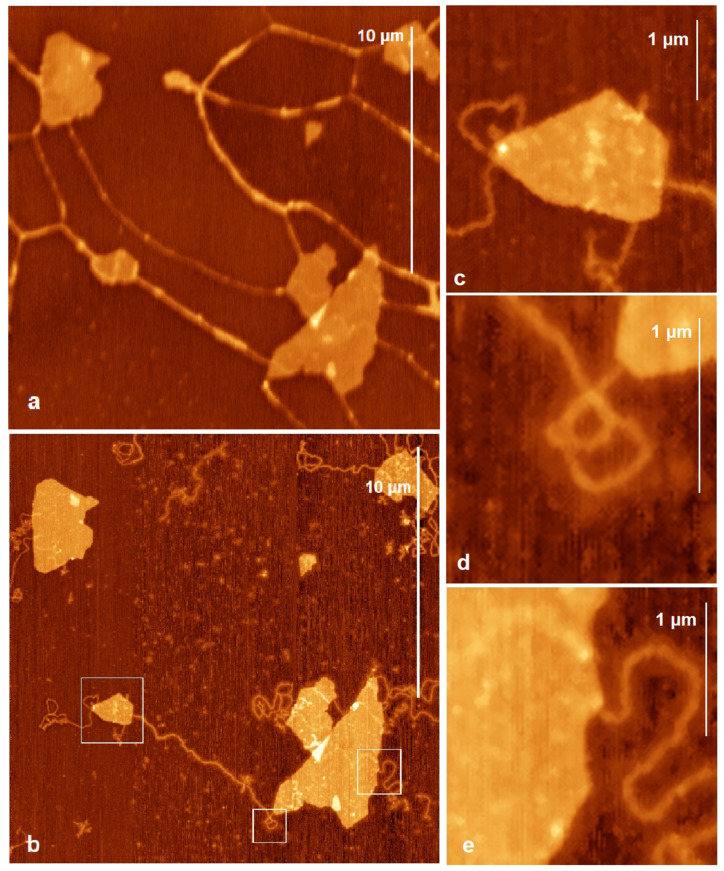
AFM images of the same area: (**a**) before and (**b**) after washing of the DNA and Mt mixture deposited on mica with MgCl_2_ solution for 1 min. (**c**–**e**) Closer view of the selected areas in (**b**) marked with white squares.

**Figure 3 ijms-24-09827-f003:**
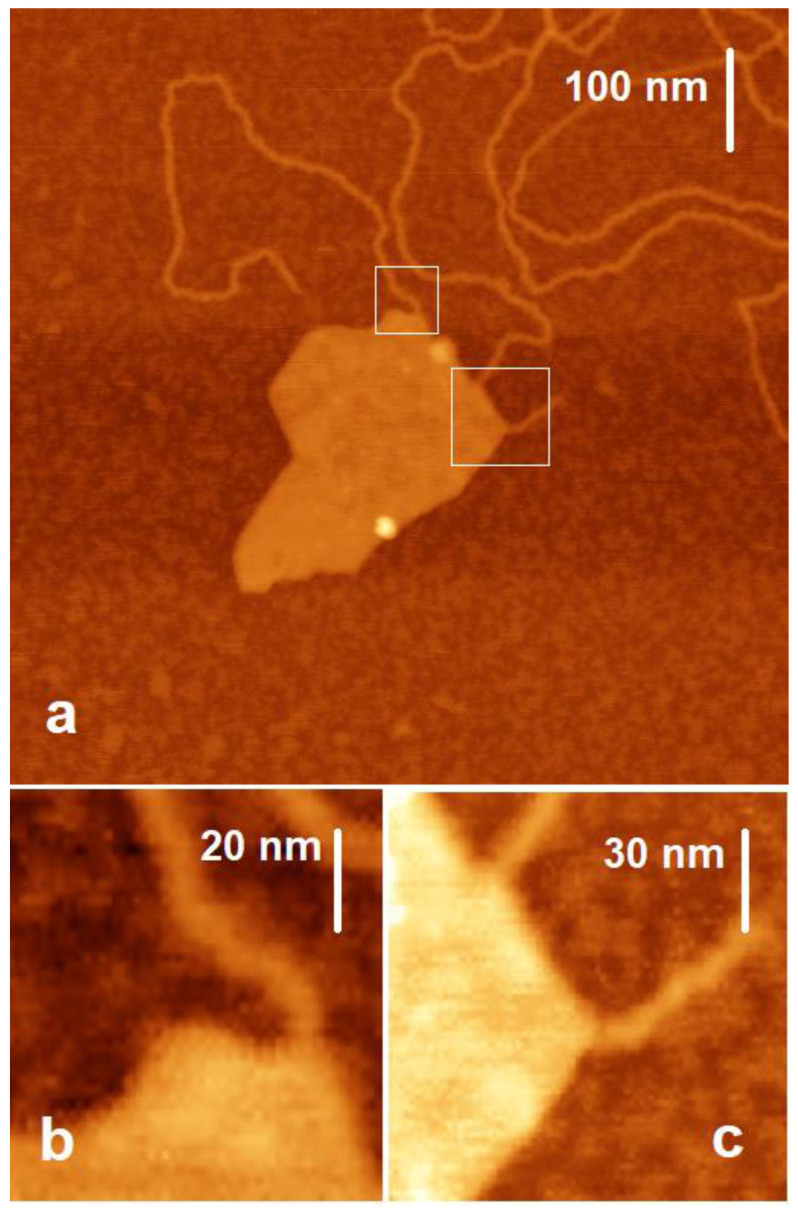
AFM images of the DNA and Mt mixed in solution with MgCl_2_ and deposited on mica (**b**,**c**). Closer view of the selected areas in (**a**) marked with white squares.

**Figure 4 ijms-24-09827-f004:**
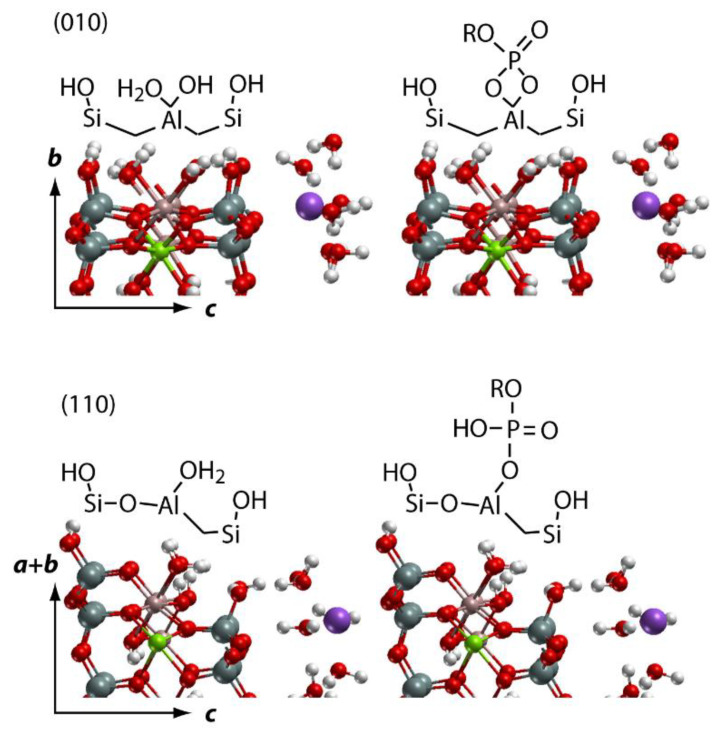
Structures of the edge surfaces of 2:1 clay minerals: (010), shown in the upper panel, and (110), shown in the lower panel. Bidentate and monodentate coordination of the phosphate group to the edge surfaces is shown schematically on the upper and lower panel, respectively. Color coding: red—O; white—H; grey—Si; brown—Al; green—Mg; violet—Na.

**Figure 5 ijms-24-09827-f005:**
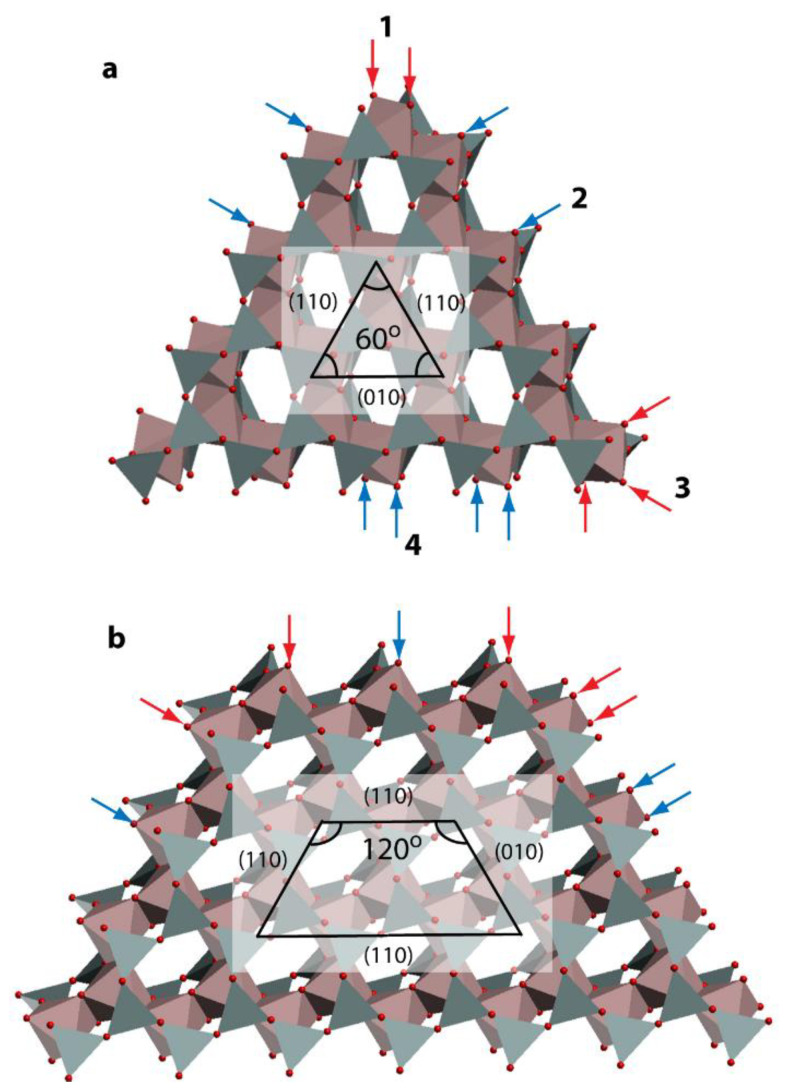
Top view of two particles formed by cutting the Mt bulk along [010] and [110] directions. (**a**) The resulting angles between the edges are all 60°. (**b**) The angles between the edges in the upper part of the particle are 120°. Arrows show various dangling surface groups. Numbers 1 to 4 indicate Al centers belonging either to edge surfaces or to their joints.

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
