# Peer review of "Features of DNA–Montmorillonite Binding Visualized by Atomic Force Microscopy"

_ijms, 2023, doi:10.3390/ijms24129827_

Round 1

Reviewer 1 Report

The manuscript “Features of DNA-montmorillonite binding visualized by atomic-force microscopy” by Kraevsky et al reports AFM imaging of Lambda DNA adsorbed on montmorillonite (Mt) nanoparticles. The manuscript is well written, presents extensive bibliography on the subject and is definitely of interest to the readers of IJMS.

The AFM images reported by the authors are similar to TEM images in Franchi et al, Orig Life Evol Biosph 372 1999, 29, 297, Figures 4 and 5 (Ref. [26] in the manuscript), however several other effects were observed, including formation of DNA bundles without Mg2+ ions, unwrapping of bundles upon addition of Mg2+ and more complex aggregates presented in Supplemental Information only. However, the importance of the images reported in the manuscript is not directly linked to bulk measurements in the references. Furthermore, several other issues remained unclear after reading the manuscript and need to be explained in more detail for the sake of readers.

1.     The authors claim that binding is mainly located along the edges Mt particles, however both bundles and single DNA molecules tend to traverse Mt particles. Moreover, as can be seen in Fig 2 a and 2 b, even after re-hydration, DNA remains on the same paths on Mt nanoparticle faces (i.e. traverses them along the same trajectory) indicating strong binding to the surfaces.

2.     The images shown in Fig. S4 seem to indicate that bundles of DNA are formed on the surface of Mt particles even in the presence of Mg2+ ions. Can the authors elaborate on this? Can the presence of bundles in between the particles can be considered as an indicator of possible particle stacking and liquid phases formation as in Ref. 55 in the manuscript?

3.     An important issue, arising when structures are prepared by consecutive addition of components to the surface, is to what extent the obtained aggregates resemble those formed in the bulk solution. Can the authors comment on this? Is, for instance, the image reported in SI as Fig. S2 obtained from bulk solution? Why pH 3.5 was selected for this image and why it is so different from Fig. S3 obtained at pH 9? What is the expected pKa of the considered binding sites on Mt?

4.     What is the mechanism of the aggregate formation? Are DNA-Mt conjugates form in solution and then adsorb on mica, or Mt particle adsorb on mica first? Can the authors show AFM images corresponding to the individual steps of their deposition process?

5.     Some minor technical issues:
-the line showing the cross-section position in Fig. 1 is too thin and barely visible.
- line 70 should probably read “… is not comprehensive without using different types of direct visualization technique”.

English language quality is Ok.

Author Response

The manuscript “Features of DNA-montmorillonite binding visualized by atomic-force microscopy” by Kraevsky et al reports AFM imaging of Lambda DNA adsorbed on montmorillonite (Mt) nanoparticles. The manuscript is well written, presents extensive bibliography on the subject and is definitely of interest to the readers of IJMS.

The AFM images reported by the authors are similar to TEM images in Franchi et al, Orig Life Evol Biosph 372 1999, 29, 297, Figures 4 and 5 (Ref. [26] in the manuscript), however several other effects were observed, including formation of DNA bundles without Mg2+ ions, unwrapping of bundles upon addition of Mg2+ and more complex aggregates presented in Supplemental Information only. However, the importance of the images reported in the manuscript is not directly linked to bulk measurements in the references. Furthermore, several other issues remained unclear after reading the manuscript and need to be explained in more detail for the sake of readers.

We would like to thank the reviewer for valuable comments.

  1. The authors claim that binding is mainly located along the edges Mt particles, however both bundles and single DNA molecules tend to traverse Mt particles. Moreover, as can be seen in Fig 2 a and 2 b, even after re-hydration, DNA remains on the same paths on Mt nanoparticle faces (i.e. traverses them along the same trajectory) indicating strong binding to the surfaces.

We would like to thank the referee for this important question. Of course, DNA can bind to the basal surface of Mt in the presence of magnesium particles. Chemically Mt and mica are very similar, but the negative surface charge of mica is stronger. That is why we use the wording “the DNA interact mostly with Mt edges”.

  1. The images shown in Fig. S4 seem to indicate that bundles of DNA are formed on the surface of Mt particles even in the presence of Mg2+ ions. Can the authors elaborate on this? Can the presence of bundles in between the particles can be considered as an indicator of possible particle stacking and liquid phases formation as in Ref. 55 in the manuscript?

The bundles shown in Fig. S4 were formed not on the surface of Mt particles, but rather between the surfaces of mica and Mt particles. DNA were probably anchored by the Mt particle and did not have time to disperse at a sufficient distance so that the AFM - probe could distinguish them separately from the single Mt sheet.

We would like to thank the referee for the nice suggestion that the formation of such bundles can explain the formation of anisotropic phases with the distance between the planes less than the DNA length, as in Ref. 55. We add this information to the text, “The presence of bundles in between the Mt particles and mica (Fig. S4) can be considered as an indicator of possible particle stacking and liquid phases formation as in [54].”

  1. An important issue, arising when structures are prepared by consecutive addition of components to the surface, is to what extent the obtained aggregates resemble those formed in the bulk solution. Can the authors comment on this? Is, for instance, the image reported in SI as Fig. S2 obtained from bulk solution? Why pH 3.5 was selected for this image and why it is so different from Fig. S3 obtained at pH 9? What is the expected pKa of the considered binding sites on Mt?

All aggregates were formed in solution near the surface. Samples were not dried at intermediate stages.

 Protonation, and hence the charge of the edge surface of Mt, depends on pH. Control experiments with different pH should have shown that in an acidic environment, DNA is better adsorbed on the edge surface than in an alkaline one. However, due to the fact that DNA coiled in an acidic environment, and the samples were overloaded with salt, this could not be confirmed by AFM.

 We have not calculated pKa of the considered binding sites on Mt. Based on work [Tournassat, C et al. 2016,  doi: 0.1021/acs.est.6b04677], we can assume that pKa should be in the range of 3-8 or lower.

  1. What is the mechanism of the aggregate formation? Are DNA-Mt conjugates form in solution and then adsorb on mica, or Mt particle adsorb on mica first? Can the authors show AFM images corresponding to the individual steps of their deposition process?

Mt particles first adsorbed on mica, and then DNA were added without drying. In this case we can observe the separate DNA-Mt conjugates. If one prepares DNA-Mt conjugates in the solution, bulky aggregates will form which will be impossible to analyze with AFM.

  1. Some minor technical issues:
    -the line showing the cross-section position in Fig. 1 is too thin and barely visible.
    - line 70 should probably read “… is not comprehensive without using different types of direct visualization technique”.

We have corrected the found mistakes.

Comments on the Quality of English Language

English language quality is Ok.

Reviewer 2 Report

The manuscript of Kraevsky and collaborators titled “Features of DNA-montmorillonite binding visualized by 2 atomic-force microscopy” investigate the structural characteristics of montmorillonite functionalised with DNA using AFM. The authors found the unfolding of duplex DNA into DNA strands when using Mg2+. This is an already well-known process in which the single stranded DNA strands are favoured at high ionic strength conditions because the stabilisation effect of ions that neutralise each phosphate group of DNA. They identified the unfolded DNA at the edges of the clay and used DFT calculations to support their findings. Overall, the work lack of enough interest to be published at IJMS and I strongly recommend to publish in a more specialised journal. In addition, I recommend to correct the English since the text contains so many typos.

Some typos are in the text and therefore, the English shall be revised.

Author Response

Comments and Suggestions for Authors

The manuscript of Kraevsky and collaborators titled “Features of DNA-montmorillonite binding visualized by 2 atomic-force microscopy” investigate the structural characteristics of montmorillonite functionalised with DNA using AFM. The authors found the unfolding of duplex DNA into DNA strands when using Mg2+. This is an already well-known process in which the single stranded DNA strands are favoured at high ionic strength conditions because the stabilisation effect of ions that neutralise each phosphate group of DNA. They identified the unfolded DNA at the edges of the clay and used DFT calculations to support their findings. Overall, the work lack of enough interest to be published at IJMS and I strongly recommend to publish in a more specialised journal. In addition, I recommend to correct the English since the text contains so many typos.

We thank the reviewer 2 for carefully reading the manuscript and the comments. We actually do not see the unzipping of double-stranded DNA into single stranded DNA and do not observe any single-stranded DNA. We have improved the wording of the section to avoid the ambiguity in terms of interpretation. Without Mg2+ we observe bundles which consist of several DNA molecules. Once Mg2+ is added we see separate dsDNA molecules but no ssDNA strands. So we could not agree with referee recomendation.

We agree with the remark about the quality of English and revised the text substantially.

Reviewer 3 Report

The manuscript entitled “Features of DNA-montmorillonite binding visualized by Atomic Force Microscopy” an interesting study on molecular level details of DNA complexed with nano-clay montmorillonite. Study reveals that the addition of Mg2+ cations lead to the separation of DNA fibers into separate molecules which bind mainly to the edge joints of the montmorillonite particles, and results in DNAs wrapping around the montmorillonite particles. The results are exciting for application reversible sorption of nucleic acids onto montmorillonite surface for both RNA and DNA isolation for further reverse transcription and polymerase chain reaction (PCR).

1.     Line 108; what does author mean “These DNA 108 fibers can reach 40 nm in width and 2 nm in height”. Figure 1c doesn’t support this.

2.     Line 143; Interestingly, the measured width of the strands in this case is about 15 nm. I think authors should clarify the difference between width and height.

3.     Line 165-170; “Again, it can be seen that in presence of magnesium ions, the DNA on the mica is unfolded into separate molecules.” Do author mean single stranded DNA?

4.     Line 238-240; authors should elaborate application part.

Minor:

1.     Line 101, The AFM technique is very

2.     Line 248, concentration g/L

3.     Line 248-249, ultrasonication parameters should be mentioned.

4.     Line 278, interacting with Mt s. Mt s should be replaced with Mt particles throughout the manuscript.

The quality of English language should be improved. There are several spelling and grammatical errors. The scientific writing part of results and discussion section is poor.   

Author Response

Comments and Suggestions for Authors

The manuscript entitled “Features of DNA-montmorillonite binding visualized by Atomic Force Microscopy” an interesting study on molecular level details of DNA complexed with nano-clay montmorillonite. Study reveals that the addition of Mg2+ cations lead to the separation of DNA fibers into separate molecules which bind mainly to the edge joints of the montmorillonite particles, and results in DNAs wrapping around the montmorillonite particles. The results are exciting for application reversible sorption of nucleic acids onto montmorillonite surface for both RNA and DNA isolation for further reverse transcription and polymerase chain reaction (PCR).

We would like to thank the reviewer for valuable comments.

  1. Line 108; what does author mean “These DNA 108 fibers can reach 40 nm in width and 2 nm in height”. Figure 1c doesn’t support this.

The profile of the fiber from Fig. 1c is not measured for the widest part but rather for a characteristic part which lies along the Mt edge. In Fig. 1b one could find parts of fibers with a larger width.

  1. Line 143; Interestingly, the measured width of the strands in this case is about 15 nm. I think authors should clarify the difference between width and height.

The question is good. Height is the measurement in the direction perpendicular to the substrate. It can be measured relatively salty surroundings or mica if the salt concentration is low. The width lateral size. We have added the phrase “because the measurement shows the value relative to the salt layer rather than the surface. The width is also smaller than for the fibers.”

  1. Line 165-170; “Again, it can be seen that in presence of magnesium ions, the DNA on the mica is unfolded into separate molecules.” Do author mean single stranded DNA?

In this case we mean double-stranded DNA. Sentence now corrected as:

“Again, it can be seen that in presence of magnesium ions, the DNA fiber on the mica is unfolded into separate double-stranded DNA molecules.”

  1. Line 238-240; authors should elaborate application part.

We would like to thank the referee for the suggestion. We have added the following

Currently silica nano- and microstructures are widely applied for industrial and household applications (food, beer production, cosmetics) as well as for clinical purposes in diagnostics and pharmaceutical products [67]. Because of their small sizes and large surface area, silica nanostructures exhibit unique bioactivities and are capable of interactions with cellular or subcellular structures. Binding of SNS and Mt with nucleic acids may be used for molecular diagnostics by means of reverse transcription with real time PCR, ligase chain reaction, loop-mediated isothermal amplification (LAMP) and molecular hybridization of nucleic acids. Importantly, DNA and RNA can be adsorbed and desorbed from the Mt surface at different osmotic conditions (see the section PCR experiment in Supplementary Material, also [20]). The biocompatible and safe nanomaterals with DNA-binding capacities are also necessary for gene delivery applications such as gene immunization and therapy[68]. Non-viral vectors have been preferred due to their ease of production, controlled chemical composition, high chemical versatility, biosafety issues, and low immunogenicity, and constituted an ideal alternative to viral vectors.

Minor:

  1. Line 101, The AFM technique is very
  2. Line 248, concentration g/L
  3. Line 248-249, ultrasonication parameters should be mentioned.
  4. Line 278, interacting with Mt s. Mt s should be replaced with Mt particles throughout the manuscript.

Thank you for the noticing this, we have corrected these points

Comments on the Quality of English Language

The quality of English language should be iauthors should elaborate application part.mproved. There are several spelling and grammatical errors. The scientific writing part of results and discussion section is poor.   

We agree with the remark about the quality of English and revised the text substantially.

Round 2

Reviewer 1 Report

I thank authors for addressing the comments. The manuscript can be accepted for publication. 

The authors are advised to thoroughly read the manuscript and do spellchecking.

Reviewer 2 Report

The work has improved substantially in the discussion of the results.

The English has greatly improved from the previous draft.